# The impact of HIV-1 subtypes on virologic and immunologic treatment outcomes at the Lagos University Teaching Hospital: A longitudinal evaluation

Ann Abiola Ogbenna[1☺]*, Seema Meloni[2☺], Seth Inzaule[3], Raph L. Hamers[3,4,5], Kim Sigaloff[3,6], Akin Osibogun[7], Titilope Adenike Adeyemo[1], Prosper Okonkwo[8], Jay Osi Samuels[8], Phyllis J. Kanki[2], Tobias F. Rinke de Wit[3], Alani Sulaimon Akanmu[1☺]

1 Department of Haematology and Blood Transfusion, Faculty of Clinical Sciences, College of Medicine, University of Lagos, Lagos, Nigeria, 2 Department of Immunology and Infectious Disease, Harvard T. H. Chan School of Public Health, Boston, Massachusetts, United States of America, 3 Department of Global Health, Amsterdam UMC, Amsterdam Institute for Global Health and Development, University of Amsterdam, Amsterdam, Netherlands, 4 Eijkman-Oxford Clinical Research Unit, Jakarta, Indonesia, 5 Centre for Tropical Medicine and Global Health, Nuffield Department of Medicine, University of Oxford, Oxford, United Kingdom, 6 Department of Internal Medicine, Division of Infectious Diseases, Amsterdam UMC, Amsterdam, Netherlands, 7 Department of Community Health and Primary Care, College of Medicine, Faculty of Clinical Sciences, University of Lagos, Lagos, Nigeria, 8 APIN Public Health Initiatives Ltd/Gte, Abuja, Nigeria

☺ These authors contributed equally to this work.
* aaogbenna@cmul.edu.ng

**Data Availability Statement:** All relevant data are within the manuscript and its Supporting Information files

## Abstract

### Introduction

HIV is a highly diverse virus with significant genetic variability which may confer biologic differences that could impact on treatment outcomes.

### Materials and methods

We studied the association between HIV subtypes and immunologic and virologic outcomes in a longitudinal cohort of 169 patients on combination antiretroviral therapy. Participants were followed up for 5 years. Demographic data, CD4 cell count and viral loads (VL) were extracted from medical records. Whole protease gene and codon 1–300 of the reverse transcriptase gene were sequenced and analysed.

### Results

Sixty-four percent of participants were females with a median age of 35 years. Twelve different subtypes were observed, the commonest being CRF 02_AG (55.0%) and subtypes G (23.1%). All subtypes showed steady rise in CD4 count and there was no difference in proportion who achieved CD4+ cell count rise of ≥100 cells/μL from baseline within 12 months' post-initiation of ART, or ≥350 cells/μL at 60 months' post-initiation. Median time to attaining a rise of ≥350 cells/μL was 24 months (6–48 months). The proportion that achieved

**Funding:** The author(s) received no specific funding for this work.

**Competing interests:** The authors have declared that no competing interests exist.

undetectable VL at month 6 and 12 post-initiation of ART were comparable across subtypes. At end of 5th year, there was no statistical difference in proportion with virologic failure.

## Conclusion

No association between HIV subtypes and immunologic or virologic response to therapy was observed, suggesting that current first-line ART may have similar efficacy across subtype predominating in South-West Nigeria.

## Introduction

HIV-1 remains a global health problem of unparalleled magnitude, with an estimated 36.9 million people living with HIV in 2017 [1]. The pandemic is dynamic, with 1.8 million new infections each year. An estimated 3.2 million Nigerians are currently estimated to be living with HIV, making it the second largest epidemic worldwide [2].

HIV-1 is a highly diverse virus due to significant genetic variability. It is classified into four groups: M (major), group O (outlier), group N (nonmajor nonoutlier), and P [3, 4]. HIV-1 group M is the most prevalent circulating group, has nine subtypes (designated A to D, F to H, and J and K), numerous circulating recombinant forms (CRFs) and multiple unique recombinant forms (URFs) [4, 5].

The distribution of HIV-1 subtypes and recombinants across the world varies and this regional diversity may have clinical implications. CRF02_AG is the fourth most prevalent subtype globally, together with subtype G remain the dominant variants observed in West Africa [6]. In Nigeria, subtypes A, B, C, D, F2, G, J, and group O have been identified along with several CRFs in varying proportions [7–10]. The distribution of HIV-1 variants in Nigeria seems to differ based on geography, as subtype G is most prevalent in the north and CRF02_AG in the south [8, 11].

There are significant sequence differences in the structural and regulatory genes of different HIV-1 subtypes and recent research suggests that the variability among HIV groups, subtypes and CRFs carry functional biological differences [12]. Subtypes have been shown in previous studies to be associated with disease progression [10–14] and mother-to-child transmission of HIV [15]. Reports on the impact of HIV subtypes on response to antiretroviral therapy vary; majority of studies which showed that subtypes have no effect on outcomes once on antiretroviral were either cross-sectional studies or longitudinal studies of 24 months or less [16–22]. However, Scherrer et al in a cohort study (1996–2009) reported an improved virologic outcome in white patients with non-B subtype particularly subtypes A and CRF02_AG compared to subtype B [21]. Resistance rates among children has been reported to be higher for non-B subtypes than for B subtypes; however, in the same study, subtypes were not associated with virologic response at 24 and 48 weeks after initiation of treatment [19]. While De Wit et al, reported no difference in the proportion of patients with viral loads below 400 copies/mL at month 24 post-initiation of ART, they found a significant difference in the median CD4+ T cell increase at month 24 when data from subtype B and non-subtype B infected patients were compared [18]. Mortality has also been reported to be associated with subtype D compared to other subtypes, though this finding may be confounded by socio-demographic factors [23, 24].

The majority of studies examining association of HIV-1 subtype with patient outcomes have largely focused on subtype B, the commonest variant in the USA and Western Europe and one that represents less than 15% of HIV-1 infections worldwide. Few studies have

evaluated the effect of HIV subtypes for periods longer than 48 weeks and these were on ARV naïve HIV infected subjects [13, 24–26]. In this study, we examined the subtype distribution and the effect of these subtypes on disease outcome over a 5-year period in a cohort of patients receiving ART in a large teaching hospital location in southwest Nigeria.

## Materials and methods

The study was an observational longitudinal study that took place in Lagos University Teaching Hospital (LUTH) at the HIV clinic, which provides care for over 8,000 HIV-positive patients. This study included all HIV-infected patients attending the APIN clinic of LUTH that were newly initiated on ART between September 2008 and June 2009, who provided informed consent, were above the age of 18 years, initiated on first-line ARV according to National guidelines and had subsequent clinical data [27]. The study was reviewed and approved by the Health Research and Ethical Committee of the Lagos University Teaching Hospital, the Institutional Review Board at the Harvard T. H. Chan, School of Public Health, the academic medical center of Amsterdam and the APIN Public Health Initiatives, Nigeria. HIV-2 positive patients and those with dual infections were excluded. The criteria for initiation of ART were the presence of a CD4+ cell count of < 350 cells/μL or the presence of symptomatic HIV disease. Medical records from the electronic medical records developed by Harvard APIN program [28] were used to obtain demographic data (gender and age), clinical and laboratory data (ART regimen at ART initiation, CD4+ cell counts and viral load values). Data for serial CD4 count and viral loads were extracted from electronic data base at baseline, then 3 monthly for the first year, subsequently every 12 months till the 60th month. This made a total of nine data points (baseline, 3-months, 6-months, 9-months, 12-months, 24-months, 36-months, 48-months and 60-months), (S1 Fig). Data for HIV sub-types was a secondary data obtained from the PASER-M study, which assessed the prevalence of primary resistance in 6 African countries after ART roll-out. One of the secondary objective of the study was to determine the relationship between HIV subtypes and ART drug resistance. The PASER-M original study design was a prospective cohort study. Its sample size was estimated at a minimum of 190 individuals per site based on virologic outcome after 24 months on ART, 20% loss to follow up and a 25% mortality rate after 24 months [29, 30]. In the LUTH site, an initial 240 patients were recruited, 198 participants had baseline sequence data. Of these, only 169 participants had subsequent clinical data and hence were included in this present study. At baseline, five mls of blood was drawn into EDTA bottles for genetic analysis. Cryopreserved plasma samples obtained before initiation of therapy were shipped on dry ice to University of Witwatersrand, South Africa for sequence analysis in 3 batches. The South African laboratory used the NucliSens EasyQ real-time Assay version 2.0 (bio-Merieux, Lyon, France) for reference HIV RNA determination. For samples with viral load >1000RNA copies/mL, the whole of the protease gene and codons 1–300 of reverse transcriptase gene were sequenced. The Laboratory used an in-house sequencing method with an ABI Prism 3730 Genetic analyzer (Applied Biosystems, Foster City CA) [30, 31]. Subtypes were determined using the REGA HIV-1 subtyping algorithm version 2.0 [32]. Additional STAR genotype analysis was carried out when required [33].

A favorable immunologic response was defined as CD4+ cell count rise of ≥100 copies/μL from baseline CD4+ cell count within 12 months, or CD4+ cell count ≥350cells/μL at end of study period. Virologic failure was defined as two consecutive HIV RNA levels >1000copies/mL following viral suppression and at least 6 months on ART. Viral suppression was defined as HIV RNA levels ≤1000 copies/mL. Viral rebound was defined as VL>1,000copies/mL following suppression.

All statistical analyses were conducted using SPSS version 21. Baseline VL and CD4+ cell counts were compared between HIV subtypes using the Kruskal-Wallis test. Proportions with an increase in CD4+ cell of > 100 cells/μL within first 12 months of initiation of ART and virologic rebound/failure were compared across subtypes using the Chi-squared test. Kaplan-Meier curves were used to examine time to CD4+ count rise to ≥350 cells/μL and VL≤1000 copies/mL after initiation of ART across subtypes, with the log-rank test being used to test the significance of observed differences between groups. Survival curves were drawn using Graph-Pad Prism version 8.4.3 (686). Median CD4+ cell rise overtime was also determined and presented as a graph.

## Results

One hundred and sixty-nine HIV-infected patients who gave consent and were newly initiated on ART as per National guidelines were included in this evaluation. They were followed for up to five years post-initiation of ART. The majority (64.3%) of participants were females and male: female ratio was comparable across subtypes. "Table 1"

The median age of participants was 35 years with majority (69.8%) being below the age of 40 years. The baseline CD4+ cell count and viral load did not statistically differ between subtypes (P>0.05). "Table 1" One hundred and three (60.9%) of participants were on AZT/3TC based regimen and 52 (38%) on TDF/FTC based regimens. "Table 1" The third drug in the ARV regimen was Nevirapine in 139 (82.2%), Efavirenz in 29 (17.2%) and Saquinavir/Ritonavir in 1 (0.6%) of participants

A complex HIV-1 diversity was seen, with multiple subtypes (D, G, J, K) and CRFs (02_AG, 01_AE, 03_AB, 14_BG, 06_cpx, 18_cpx, 36_cpx, 43_02G). The most common variants were CRF 02_AG (55.03%) and G (23.67%). "Table 1" CRF06_cpx and CRF18_cpx accounted for 9.47% and 6.51% respectively. The remaining variants accounted for only 5.33% of the sequence diversity. These was made up of subtypes D (1.18%), J (0.59%), K (0.59%), CRF01_AE (0.59%), CRF03_AB (0.59%), CRF14_BG (0.59%), CRF36_cpx (0.59%), and CRF43_02G (0.59%). "Table 1"

At 12 months' post-initiation of ART, the proportion of participants who achieved favourable immunologic response was comparable across subtypes "Table 2".

The proportion that achieved an virologic suppression at month 3, 6 and 12 post-initiation of ART were also comparable among the different subtype populations. "Table 3"

All subtypes showed a steady rise in CD4 + cell count; however, CFR06_cpx and CRF18_cpx demonstrated frequent peaks and dips. In both these subtypes, a more detailed scrutiny of data revealed that both had one participant with inconsistently high CD4 counts which corresponded to the peaks in the graph. The few number of participant in both populations made the effect of this "outliers" marked. "Fig 1".

At the end of the observation period, there were no significant differences in proportion of patients with viral suppression or with CD4 counts≥350cells/μL. "Table 4".

The median time to attaining CD4+ cell count increase of ≥350cells/μL was 24 months (IQR: 6–48 months). The shortest median time to CD4+ cell count increase of ≥350cells/μL was observed in CRF 06_cpx-infected population (12 months; 95% confidence interval [CI]: 0.00–44.93 months) compared to CRF 18_cpx (24 months; 95% CI, 9.80–38.20 months), CRF 02_AG (24 months; 95% CI: 18.77–29.23 months) and subtype G (36 months; 95% CI: 22.90–49.10 months). However, this differences in median time to a rise in CD4+ cell count of ≥350cells/μL were not statistically significant. (p>0.05) No significant difference in the survival curves for the different subtype population was observed, P>0.05 "Fig 2".

**Table 1. Baseline demographics and clinical characteristics of the study population by subtypes.**

| | HIV subtype N (%) | | | | | | P value |
|---|---|---|---|---|---|---|---|
| | G | CRF02AG | CRF06 cpx | CRF18 cpx | Others | Total | |
| **Frequency (%)** | **40 (23.7)** | **93 (55.0)** | **16 (9.5)** | **11 (6.5)** | **9 (5.3)** | **169 (100)** | |
| **Sex** | | | | | | | |
| Male | 11 (27.5) | 37 (39.8) | 8 (50.0) | 2 (18.2) | 3 (33.3) | 61 (36.1) | 0.32 |
| Female | 29 (72.5) | 56 (60.2) | 8 (50.0) | 9 (81.8) | 6 (66.7) | 108 (63.9) | |
| **Median Age**(years) | 34 (30–38) | 35 (32–43) | 38 (35–46) | 35 (33–41) | 43 (35–48) | 35 (32–42.5) | 0.05 |
| **Median CD+4 cells/μL(IQR)** | 128 (62.0–185.8) | 127 (64.0–198.0) | 146 (81.3–202.3) | 144 (121.8–211.0) | 116 (93.5–187.5) | 131 (66.3–194) | 0.81 |
| **Median VL Log copies/mL(IQR)** | 5.24 (4.89–5.60) | 4.87 (4.09–5.48) | 5.27 (4.28–5.64) | 4.91 (3.40–5.75) | 5.47 (5.02–5.86) | 5.06 (4.29–5.59) | 0.16 |
| **ARV at initiation**[*] | | | | | | | |
| AZT/3TC | 24 (60) | 54 (58.0) | 12 (75) | 6 (54.5) | 7 (77.8) | 103 (60.9) | |
| TDF/FTC | 14 (35) | 31 (33.3) | 2 (12.5) | 3 (27.3) | 2 (22.2) | 52 (30.8) | |
| ABC/3TC | 2 (5) | 6 (6.5) | 2 (12.5) | 1 (9.1) | 0 (0) | 11 (6.5) | |
| D4T/3TC | 0 (0) | 2 (2.2) | 0 (0) | 1 (9.1) | 0 (0) | 3 (1.8) | |

IQR-interquartile range, AZT-Zidovudine, 3TC-Lamivudine, TDF- Tenovofir, D4T-Stavudine.

[*]ARV are classified based on NRTI backbone.

The baseline VL was comparable across the different subtype populations "Table 1". Eight of 15 (53.3%) of subjects with CRF 06_cpx had virologic rebound during the study. This was significantly higher than those with virologic rebound among subjects with subtype G, 11/18 (37.9%) and subtype 18_cpx, 2/ 8 (20%). P <0.05. However, there was no statistically significant difference in the proportion who had virologic failure during the study period. Overall, 8 (6.1%) of study population had a virologic failure. "Table 5".

The shortest median time to VL ≤ 1,000 copies/mL were observed in Subtype G infected population (3months; 95% confidence interval [CI], 2.22–3.78 months). Other major subtypes had a median time of 6 months to VL ≤ 1,000 copies/mL with CI as follows; CRF 06_cpx (3.99–8.01 months), CRF 18_cpx (1.48–10.52 months) and CRF 02_AG (4.91–7.09 months). The overall median time to VL ≤1,000 copies/mL for the study population was 3 months (CI, 2.20–3.81 months). However, there was no significant difference in the survival curves for the different subtype populations as P>0.05. "Fig 3"

A hundred and eleven (65.7%) participants were still in care, 54 (34%) were lost to follow up, 2 (1.2%) had died, and 2 (1.2%) had been transferred to another facility "Table 4". Participants with CRF 06_cpx and CRF 18_cpx had a significantly greater proportion still in care compared to the other subtypes. P<0.05 "Table 4".

**Table 2. Proportion with favorable immunologic response (CD4+ cell count rise of ≥100 copies/μL from baseline) at 12 months after initiation of ARV drugs.**

| | HIV subtypes n (%) | | | | | |
|---|---|---|---|---|---|---|
| | G | 02_AG | 06_cpx | 18_cpx | Others | Total |
| **CD 4+ cell count rise at 12 months** | | | | | | |
| ≥100 cells/μL | 59 (91.7) | 59 (96.7) | 11 (91.7) | 10 (100) | 8 (100) | 110 (97.6) |
| < 100 cells/μL | 2 (8.3) | 2 (3.3) | 1 (8.3) | 0 (0) | 0 (0) | 5 (4.3) |

P>0.05.

**Table 3. Proportion with viral suppression (VL ≤1000 copies/mL)at 3 months, 6 months and 12 months after initiation of ARV drugs.**

|  | G | 02_AG | 06_cpx | 18_cpx | Others | Total |
|---|---|---|---|---|---|---|
| **VL at 3 months (copies/mL)** |  |  |  |  |  |  |
| >1000 | 2 (8.7) | 11 (25.6) | 2 (25.0) | 2 (40.0) | 1 (14.3) | 18 (20.9) |
| ≤ 1000 | 21 (91.3) | 32 (74.4) | 6 (75.0) | 3 (60.0) | 6 (85.7) | 68 (79.1) |
| **VL at 6 months (copies/mL)** |  |  |  |  |  |  |
| >1000 | 3 (17.6) | 8 (18.2) | 1 (11.1) | 1 (16.7) | 2 (28.6) | 15 (18.1) |
| ≤ 1000 | 14 (82.4) | 36 (81.8) | 8 (88.9) | 5 (83.3) | 5 (71.4) | 68 (81.9) |
| **VL at 12 months (copies/mL)** |  |  |  |  |  |  |
| >1000 | 3 (13.0) | 9 (15.8) | 3 (27.3) | 1 (10.0) | 2 (25.0) | 18 (16.5) |
| ≤ 1000 | 20 (87.0) | 48 (84.2) | 8 (72.7) | 9 (90.0) | 6 (75.0) | 91 (83.5) |

P>0.05.

## Discussion

In this evaluation of HIV-1 genetic diversity in southwest Nigeria, we found the most common variants to be CRF02_AG and subtype G, which is similar to what previous studies examining genetic diversity in Nigeria have found [7, 8, 10, 11, 34] The dominant spread of CRF02_AG in West Africa has been attributed to the replicative fitness it confers over subtype A and G in the same geographical region [35]. The other subtypes, including D, J, K, CRF43_02G, CRF06_cpx and CRF36_cpx, which were also found in this study have also been reported in other West African settings, largely at a lower prevalence [36]. Reported prevalence of CRF06_cpx in Nigeria vary between 4.4% and 11% [8, 37]. in this study its prevalence was 9.5%. The prevalence of CRF43_02G has been reported as higher in Abuja (18.5%) than what we found in Lagos (0.59%) [38]. CRF14_BG and CRF03_AB, which have not previously been reported in Nigeria account for 0.59% each of subtypes report in this study.

To date, most reports of associations between subtypes and immunologic outcomes have focused on comparing patients infected with subtype B to those with non-subtype B viruses. As such, different studies lump different subtypes as non-subtype B, making it difficult to know the impact of less predominant subtypes. Subtype B is uncommon in Nigeria. In this

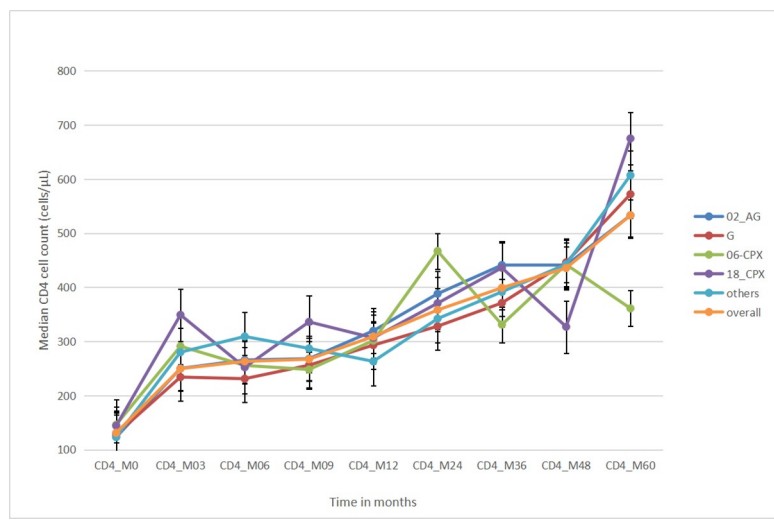

**Fig 1. Median CD4+ cell count rise over time across subtypes.**

**Table 4. Outcome at 60th months of follow-up of each patient from time of initiation of ARV.**

| | HIV subtypes n (%) | | | | | |
|---|---|---|---|---|---|---|
| | G | 02_AG | 06_cpx | 18_cpx | Others | Total |
| Status in care* | | | | | | |
| Still in care | 26 (65.0) | 53 (57.0) | 13 (81.3) | 10 (90.9) | 9(100) | 111 (65.7) |
| LTFU | 13 (32.5) | 38 (40.8) | 2 (12.5) | 1 (9.1) | 0 (0) | 54 (32.0)) |
| Dead | 0 (0) | 1 (1.1) | 1 (6.3) | 0 (0) | 0 (0) | 2 (1.2) |
| Transferred to other facilities | 1 (2.5) | 1 (1.1) | 0 (0) | 0 (0) | 0 (0) | 2 (1.2) |
| Immunologic status** | | | | | | |
| Median CD4+(cells/μL) | | | | | | |
| < 350 | 3 (30) | 3 (20.8) | 2 (40) | 1 (33.3) | 1 (25) | 12(26.1) |
| ≥ 350 | 7 (70) | 19 (79.2) | 3 (60) | 2 (66.7) | 3 (75) | 34 (73.9) |
| Virologic status** | | | | | | |
| Median VL (copies/mL) | | | | | | |
| < 1000 | 2 (66.7) | 12 (92.3) | 5 (100) | 0 (0) | 1 (100) | 20 (90.9) |
| ≥ 1000 | 1 (33.3) | 1 (7.7) | 0 (0) | 0 (0) | 0 (0) | 2 (9.1) |

*P value (likelihood ratio) = 0.046.

**P >0.05. Only 46 (41.4%) and 22 (19.8%) of those still in care had CD4 + cell count or VL respectively done at the 60[th] month.

study, baseline CD4+ cell count and proportion who achieved favourable immunologic responses at 12 months were comparable across subtypes "Tables 1–3", similar to other studies [39, 40]. It must be noted that in a study by Geretti et al, the authors reported higher baseline CD4+ cell counts in subtype B infected patients as compared to other patients that were maintained throughout the 39 weeks duration of study [41].

A study in Malaysia reported a shorter median time for CD4+ T-cell count increase to 350 cells/μL for CRF01_AE compared to subtype B-infected patients [39]. However, in two studies from China, where the CRF01_AE accounts for 50–60% of HIV-1 subtypes, CRF01_AE subtype was correlated with a significant risk of accelerated HIV/AIDS progression compared to non-CRF01_AE subtypes [40–42]. Similar rates of CD4+ cell count recovery for all subtypes as documented in this study has been reported in several other studies [40, 41, 43]. However, a study in France by Chaix M et al showed that patients infected with a non-B virus including CRF02_AG, had better immunological responses between the first 18 months than those infected with a subtype-B virus [44]. In our study, sub-analysis comparing immunologic response in subtypes- CFR02_AG and other subtypes did not show a significant difference in proportion who achieved a rise in CD4+ cell count of ≥100 cells/μL at 12 months or who had a rise of ≥ 350 cells/μL at 60 months.

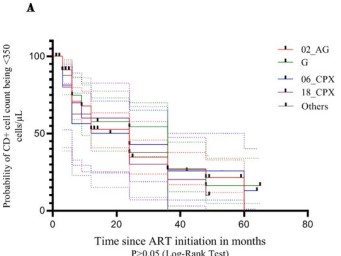 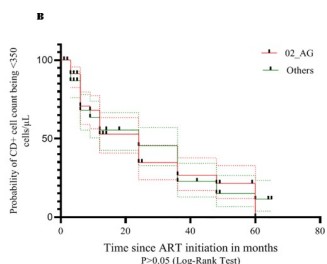

**Fig 2.** Kaplan_Meier analysis of time to reach CD4+ T_cell count of≥350 cells/μL on cART (A) Comparing the all subtypes. (B) Comparing subtypes G and CRF02_AG.

**Table 5. Proportion with virologic rebound or virologic failure during the study period.**

| | HIV subtypes n (%) | | | | | |
|---|---|---|---|---|---|---|
| | G | 02_AG | 06_cpx | 18_cpx | Others | Total |
| Virologic rebound* | | | | | | |
| No | 18 (62.1) | 56 (81.2) | 7 (46.7) | 8 (80.0) | 5 (55.6) | 94 (71.2) |
| Yes | 11 (37.9) | 13 (18.8) | 8 (53.3) | 2 (20.0) | 4 (44.4) | 38 (28.8) |
| Virologic Failure** | | | | | | |
| No | 28 (100.0) | 63 (91.3) | 15 (100.0) | 9 (90.0) | 8 (88.9) | 123 (93.9) |
| Yes | 0 (0) | 6 (8.7) | 0 (0) | 1 (10.0) | 1 (11.1) | 8 (6.1) |

*P = 0.035.

**P>0.05 Virologic failure was defined as two consecutive HIV RNA levels >1000copies/mL following viral suppression and at least 6 months on ART. Viral rebound was defined as VL>1,000copies/mL following suppression.

Though the shortest median time to VL ≤1,000 copies/mL was observed in the subtype G infected population, overall virologic outcomes were comparable across subtypes. Similar findings have been reported in several other studies [19, 41, 43–46].

Data quality is highly dependent on the completeness of clinical and laboratory values. As expected of cohort studies, this study being an observational programme study had some missing laboratory data which increased as study continued over time. At the end of the study period of this present analysis, the percentage loss to follow-up was 32.0% (Table 4). Another limitation of this study is the fact that the data on HIV subtypes was a secondary data of all available complete data as sample size was not previously calculated. The fact that VL was not assessed at all points for all patients creates the possibility of potential bias. Being an observational study, estimates on time to virologic suppression are also based only on patients with available data, so not all patients can be assessed at all time points. However, the long follow up period and the within-program comparison of subtypes which limits expected confounders are the strengths of this study.

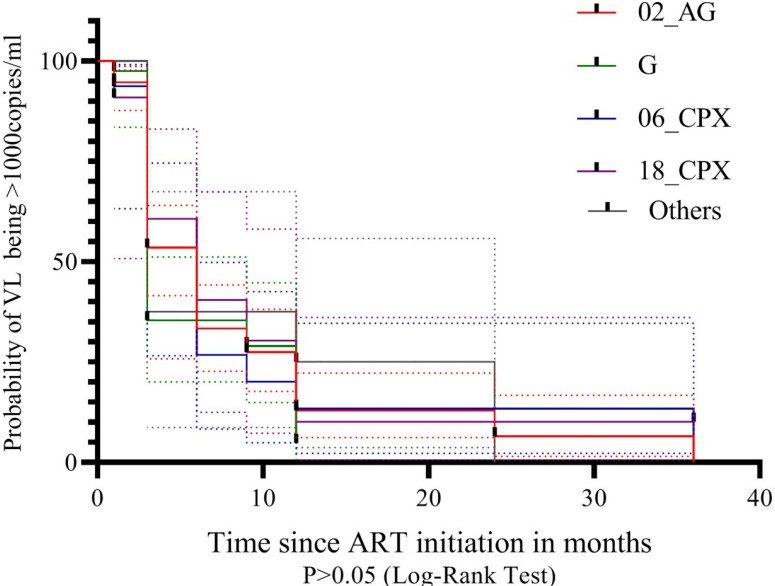

**Fig 3. Kaplan-Meier analysis of time to reach viral load of ≤ 1000 copies /mL on cART.**

## Conclusion

In conclusion, we found no evidence of an association between subtype and immunologic or virologic response to therapy, suggesting that current antiretroviral agents that are broadly in use have similar efficacy across subtypes that predominate in southwest Nigeria. The high number of HIV-1 subtypes and recombinant viruses observed in this study confirms that continuous molecular and virologic monitoring of HIV-1 in Nigeria remains of great importance.

## Supporting information

**S1 Fig. Study profile showing number of participants included in analysis at each time line.**
(DOCX)

## Acknowledgments

We would like to acknowledge and thank all patients and staff in the AIDS Prevention Initiative in Nigeria Plus (APIN Plus) Harvard PEPFAR program.

## Author Contributions

**Conceptualization:** Ann Abiola Ogbenna, Jay Osi Samuels, Tobias F. Rinke de Wit, Alani Sulaimon Akanmu.

**Data curation:** Ann Abiola Ogbenna, Seema Meloni, Alani Sulaimon Akanmu.

**Formal analysis:** Ann Abiola Ogbenna, Seema Meloni.

**Investigation:** Alani Sulaimon Akanmu.

**Methodology:** Ann Abiola Ogbenna, Alani Sulaimon Akanmu.

**Project administration:** Alani Sulaimon Akanmu.

**Supervision:** Alani Sulaimon Akanmu.

**Writing – original draft:** Ann Abiola Ogbenna, Seema Meloni, Akin Osibogun, Phyllis J. Kanki, Alani Sulaimon Akanmu.

**Writing – review & editing:** Ann Abiola Ogbenna, Seema Meloni, Seth Inzaule, Raph L. Hamers, Kim Sigaloff, Akin Osibogun, Titilope Adenike Adeyemo, Prosper Okonkwo, Jay Osi Samuels, Phyllis J. Kanki, Tobias F. Rinke de Wit, Alani Sulaimon Akanmu.

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
