## [Decision Letter · Decision Letter 0]

14 Apr 2020

PONE-D-19-29623

The Impact of HIV-1 Subtypes on Virologic and Immunologic Treatment Outcomes at the Lagos University Teaching Hospital: A longitudinal evaluation

PLOS ONE

Dear Dr. Ogbenna,

Thank you for submitting your manuscript to PLOS ONE. After careful consideration, we feel that it has merit but does not fully meet PLOS ONE’s publication criteria as it currently stands. Therefore, we invite you to submit a revised version of the manuscript that addresses the points raised during the review process.

We would appreciate receiving your revised manuscript by May 29 2020 11:59PM. To enhance the reproducibility of your results, we recommend that if applicable you deposit your laboratory protocols in protocols.io, where a protocol can be assigned its own identifier (DOI) such that it can be cited independently in the future. For instructions see: http://journals.plos.org/plosone/s/submission-guidelines#loc-laboratory-protocols

We look forward to receiving your revised manuscript.

Kind regards,

Jason Blackard, PhD

Academic Editor

PLOS ONE

Additional Editor Comments (if provided):

This is a longitudinal study of HIV disease progression based on the HIV subtype in Nigeria.  Well-designed studies evaluating the impact of HIV subtypes on disease progression are needed, particularly in resource-limited settings.

It is unclear if the 169 participants included represent all patients at this hospital that met the enrollment criteria or a subset of those individuals.  How representative are these 169 individuals of the HIV-positive population at this hospital?  In Nigeria?

A more robust discussion of the potential limitations is warranted.

The authors should who the sequence data as a phylogenetic tree.

 How do the authors know that a sufficient sample size was included to robustly evaluate if subtypes differ in disease progression?

2. Please provide additional details regarding participant consent. In the ethics statement in the Methods and online submission information, please ensure that you have specified whether consent was written or verbal/oral. If consent was verbal/oral, please specify: 1) whether the ethics committee approved the verbal/oral consent procedure, 2) why written consent could not be obtained, and 3) how verbal/oral consent was recorded. If your study included minors, please state whether you obtained consent from parents or guardians in these cases.

Reviewers' comments:

Reviewer's Responses to Questions

**Comments to the Author**

1. Is the manuscript technically sound, and do the data support the conclusions?

Reviewer #1: Yes

Reviewer #2: Yes

2. Has the statistical analysis been performed appropriately and rigorously? 

Reviewer #1: Yes

Reviewer #2: Yes

3. Have the authors made all data underlying the findings in their manuscript fully available?

Reviewer #1: Yes

Reviewer #2: No

4. Is the manuscript presented in an intelligible fashion and written in standard English?

Reviewer #1: Yes

Reviewer #2: Yes

5. Review Comments to the Author

Reviewer #1: This study is of great significance in the field of HIV treatment and monitoring of circulating subtypes, the results can be used for clinical purposes and for the treatment and care of those infected with HIV. It is also important to note the differences in subtypes circulating in developing countries such as USA and Europe with those obtained in Africa where the HIV pandemic hit the hardest. The study was well written and the author showed understanding of the subject and the methodology used was suited to this study. However, there were some issues with the reporting of the results. It was not easy to follow. This is understandable as they needed to summarize 5 years of work in a short manuscript.

• Minor issues:

1. Methods: The time points need to be mentioned here and this will help with the reporting of the time line in the results section. There is also no mention of how many samples were shipped to Johannesburg for testing by each time point. Page 5

2. Results:

o The results were very confusing to follow and understand, but this should be sorted once the time points are clearly stated in the methods section. I suggest that flow diagram showing sample availability at each of the tested time points be included and reasons for samples missing and those with missing data (Page 7). For example, in Table 3 the researchers list possible reasons why samples are missing but even in those that were still under care, the numbers reported don’t add up to the totals in the table (Page 9).

o Separate reporting of results for viral load and CD4 counts it will be clearer when reporting in the results section. Also report them by time points so that it is clear to see the rise in CD4 or viral load by time point tested. (Page 9 and 10)

o There also needs to be consistency in the number of visits, in Figure 1 there are 9 time points for CD4 count while Table 2 and 3 show that you had 2 visits (Page 8 and 9). For viral load there are 4 visits as shown in Table 2 and Table 3 (Page 8 and Page 9).

3. Discussion: there is inconsistencies in the CD4 counts of those infected with CRF 06-cpx and CRF 18-cpx subtypes (Figure 1). The authors did not show why this was happening and this happened for just these two subtype. The low numbers that they mentioned in the results section does not explain such (Page 9).

Minor issues:

1. Reference #14 is not in the text but is stated in the reference list.

Reviewer #2: In this manuscript, Ogbenna and colleagues longitudinally followed up 169 study participants infected with different subtypes who were started on antiretroviral therapy in Nigeria. The authors report that the individuals did not differ in baseline CD4+ T cell counts or viral loads according to HIV-1 subtype, and there were no differences in immunological and virological outcomes following initiation of antiretroviral therapy. Overall, this is a sound study although I found the report generally poorly written and sometimes confusing.

1) Do the authors have CD4% and CD4:CD8 ratios for participants in the cohort? Others have reported that CD4% and CD4:CD8 ratios do not normalize even after long-term antiretroviral therapy in other cohorts so it would be interesting to see whether that holds true for this cohort and if there are differences in these measures by HIV-1 subtype.

2) In table 1, only two antiretroviral drugs are shown for participants in different categories. Were participants in this cohort on two-drug regimens instead of three? Why is that the case?

3) In all the tables, there is repetition in data presentation- why for example show the numbers of both males and females? Why not just show one category (with percentage) since for these binary variables it is obvious what the numbers and percentages will be for the other category? The same applies for measures such as CD4+ T cell count increases above 100 cells/µl, viral load suppression below 1,000 copies/ml, etc.

4) The authors should show the confidence intervals for the data presented in figures 1-3. Also is this the same data in tables 1-4 and only presented differently? It would be preferable that there is no data repetition unless there is a specific separate point that the authors want to highlight by presenting the same data in different formats.

5) The figure legends for figures 1 to 3 are mixed up with the manuscript results narrative. The figure legends should be separated from the results narrative to make it easier to follow.

6) Although the REGA tool was used for subtyping, a more robust phylogenetic analysis with reference sequences would have been useful, particularly considering the extreme diversity of viral strains in Nigeria.

7) The conclusion that a higher proportion of CRF-06-cpx and CRF-18-cpx remained in care at 60 months seems tenuous and implies that there are differences between subtypes in terms of response to treatment. Unless the authors can show that this is a solid finding backed by systematic data rather than a chance finding, it should be removed from the abstract.

6. PLOS authors have the option to publish the peer review history of their article (what does this mean?). If published, this will include your full peer review and any attached files.

Reviewer #1: No

Reviewer #2: No

---

## [Author Response · Author response to Decision Letter 0]

18 Jun 2020

Academic Reviewer:

t is unclear if the 169 participants included represent all patients at this hospital that met the enrollment criteria or a subset of those individuals. How representative are these 169 individuals of the HIV-positive population at this hospital? In Nigeria?

Response: 240 adult patients aged >18 years were recruited over a period of one year. Of these, 198 participants had a baseline sequence data out of which only 169 had clinical follow up data to interrogate the impact of subtypes on treatment outcomes.

A detailed flow chart has been included as supplementary data.

(Page 5,Lines 6, 108; Page 6, )116-127

A more robust discussion of the potential limitations is warranted.

Response:

In line with the Editor’s review, we have revised and elaborated on the limitations of the study. (Page 14, Lines 249-253)

The authors should who the sequence data as a phylogenetic tree.

Response:

This present study is a secondary data analysis. We have now described the primary study in the methodology section. Reference of the primary study was also included. (Page 6, Line 119-125)

 How do the authors know that a sufficient sample size was included to robustly evaluate if subtypes differ in disease progression?

Response:This was a programme study and baseline sequence data was conducted for the participants. Our study is a secondary data analysis of all available complete data. Though the primary study for the sequence data was a prospective cohort study and appropriate sample size was calculated based on: virologic outcomes after 24 months on ART, a 20% loss to follow up and a 25% mortality rate after 24month. The fact that sample size was not calculated initially to include a 5-year follow up remains a limitation of the study. This has been alluded to in page 14, line 252-259.

Please provide additional details regarding participant consent. In the ethics statement in the Methods and online submission information, please ensure that you have specified whether consent was written or verbal/oral. If consent was verbal/oral, please specify: 1) whether the ethics committee approved the verbal/oral consent procedure, 2) why written consent could not be obtained, and 3) how verbal/oral consent was recorded. If your study included minors, please state whether you obtained consent from parents or guardians in these cases.

Response: A written consent was obtained for all participants. 

Reviewer 1

 Methods: The time points need to be mentioned here and this will help with the reporting of the time line in the results section. There is also no mention of how many samples were shipped to Johannesburg for testing by each time point. Page 5

Response:Data for serial CD4 count and viral loads were extracted from electronic data base at baseline, 3/12, 6/12, 9/12, 12 months, 24 months, 36months, 48 months and 60month. This has been reflected in the manuscript

Samples were shipped in 3 batches. This also has been reflected in the methodology(Page 6, Lines 116-119)

2. Results:

o The results were very confusing to follow and understand, but this should be sorted once the time points are clearly stated in the methods section. I suggest that flow diagram showing sample availability at each of the tested time points be included and reasons for samples missing and those with missing data (Page 7). For example, in Table 3 the researchers list possible reasons why samples are missing but even in those that were still under care, the numbers reported don’t add up to the totals in the table (Page 9).

o Separate reporting of results for viral load and CD4 counts it will be clearer when reporting in the results section. Also report them by time points so that it is clear to see the rise in CD4 or viral load by time point tested. (Page 9 and 10)

o There also needs to be consistency in the number of visits, in Figure 1 there are 9 time points for CD4 count while Table 2 and 3 show that you had 2 visits (Page 8 and 9). For viral load there are 4 visits as shown in Table 2 and Table 3 (Page 8 and Page 9).

Response:Time points have been stated in methodology andFlow chart added as a supplementary data.

Data on those still in care was obtained from drug pick up data. However, it was not possible to determine why these patients had missing laboratory results as such records are not kept in the clinic.

Table 2 has been separated for VL and CD+ count and reported by timelines. A favorable immunologic response was defined as CD4+ cell count rise of ≥100 copies/µL from baseline CD4+ cell count within 12 months, hence table 2a shows this. Viral suppression was defined as HIV RNA levels ≤ 1000 copies/mL. This is reflected in table 2b and the title has been changed to reflect the content. (Page 8 and 9)

Table 2, shows as title reflects, outcome at first 12 months hence the fewer number of visits, while figure 1 shows outcome over the whole period of study hence 9 time points.

Table 3, as title reflects shows outcome at end point only i.e. month 60

3. Discussion: there is inconsistencies in the CD4 counts of those infected with CRF 06-cpx and CRF 18-cpx subtypes (Figure 1). The authors did not show why this was happening and this happened for just these two subtype. The low numbers that they mentioned in the results section does not explain such (Page 9).

Response:In line with the Reviewer’s comment, we have revised and elaborated on probable reasons. (Page 9 Line 176-179)

Minor issues:

1. Reference #14 is not in the text but is stated in the reference list.

Response: This has been corrected in Page 4 line 84.

Reviewer 2:

 Do the authors have CD4% and CD4:CD8 ratios for participants in the cohort? Others have reported that CD4% and CD4:CD8 ratios do not normalize even after long-term antiretroviral therapy in other cohorts so it would be interesting to see whether that holds true for this cohort and if there are differences in these measures by HIV-1 subtype.

Response: No, we do not have CD % or CD4: CD8 ratio.

2) In table 1, only two antiretroviral drugs are shown for participants in different categories. Were participants in this cohort on two-drug regimens instead of three? Why is that the case?

Response:No they were on 3 drugs. however, they were classified based on NRTI backbone. (Page 6, Line 156)

3) In all the tables, there is repetition in data presentation- why for example show the numbers of both males and females? Why not just show one category (with percentage) since for these binary variables it is obvious what the numbers and percentages will be for the other category? The same applies for measures such as CD4+ T cell count increases above 100 cells/µl, viral load suppression below 1,000 copies/ml, etc.

Response: There is no repetition of data. Table titles have been edited to represent content. (Table 2a: Page 8;Table 2b: Page 9)

4) The authors should show the confidence intervals for the data presented in figures 1-3. Also is this the same data in tables 1-4 and only presented differently? It would be preferable that there is no data repetition unless there is a specific separate point that the authors want to highlight by presenting the same data in different formats.

Response: This is not the same data.Figure 1 shows the median CD4+ cell count over the whole period of study. Table 2a show proportion with a favourable immunologic response at 12 months after initiation of therapy, while table 2b shows proportion with viral suppression at variable times within the first 12 months. Table 3 shows outcome at end of study period which is 60th month. Confidence Intervals were reported as prose page 10, 176-182.

5) The figure legends for figures 1 to 3 are mixed up with the manuscript results narrative. The figure legends should be separated from the results narrative to make it easier to follow.

Response: This has been rectified. (Page 8, line 169)

The Journal requires that figure legend be placed at place where it first mentioned in article, though it is uploaded separately.

6) Although the REGA tool was used for subtyping, a more robust phylogenetic analysis with reference sequences would have been useful, particularly considering the extreme diversity of viral strains in Nigeria.

Response: As explained above, the data sequence data was a secondary data. This has been stated and referenced in the methodology.

7) The conclusion that a higher proportion of CRF-06-cpx and CRF-18-cpx remained in care at 60 months seems tenuous and implies that there are differences between subtypes in terms of response to treatment. Unless the authors can show that this is a solid finding backed by systematic data rather than a chance finding, it should be removed from the abstract.

Response: This has been removed from the abstract. (Page 2, Lines 44 & 45)

---

## [Decision Letter · Decision Letter 1]

9 Jul 2020

PONE-D-19-29623R1

The impact of HIV-1 subtypes on virologic and immunologic treatment outcomes at the Lagos university teaching hospital: A longitudinal evaluation

PLOS ONE

Dear Dr. Ogbenna,

Thank you for submitting your manuscript to PLOS ONE. After careful consideration, we feel that it has merit but does not fully meet PLOS ONE’s publication criteria as it currently stands. Therefore, we invite you to submit a revised version of the manuscript that addresses the points raised during the review process.

Please make the minor updates requested by Reviewer #2 prior to acceptance of your manuscript.

We look forward to receiving your revised manuscript.

Kind regards,

Jason Blackard, PhD

Academic Editor

PLOS ONE

Additional Editor Comments (if provided):

Please make the minor updates requested by Reviewer #2 prior to acceptance of your manuscript.

Reviewers' comments:

Reviewer's Responses to Questions

**Comments to the Author**

1. If the authors have adequately addressed your comments raised in a previous round of review and you feel that this manuscript is now acceptable for publication, you may indicate that here to bypass the “Comments to the Author” section, enter your conflict of interest statement in the “Confidential to Editor” section, and submit your "Accept" recommendation.

Reviewer #2: (No Response)

2. Is the manuscript technically sound, and do the data support the conclusions?

Reviewer #2: Yes

3. Has the statistical analysis been performed appropriately and rigorously? 

Reviewer #2: Yes

4. Have the authors made all data underlying the findings in their manuscript fully available?

Reviewer #2: Yes

5. Is the manuscript presented in an intelligible fashion and written in standard English?

Reviewer #2: Yes

6. Review Comments to the Author

Reviewer #2: In this version of the manuscript, Ogbenna and colleagues have addressed most of the concerns from reviewers. I have a few minor issues that the authors should address.

1) The authors state that antiretroviral drug categories shown are classified based on the NRTI backbone. I think it is still important to indicate what the third drug used was even if it was not uniform across all participant groups.

2) I understand that the confidence intervals for the data shown in figures 1-3 are described in the narrative. Why are they not shown in the figures?

3) The supplementary figure shown indicated that there were 169 study participants, 86 with viral load and 88 without viral load. These numbers should be checked as they do not add up to 169.

7. PLOS authors have the option to publish the peer review history of their article (what does this mean?). If published, this will include your full peer review and any attached files.

Reviewer #2: No

---

## [Author Response · Author response to Decision Letter 1]

6 Aug 2020

Reviewer 2

Thank you Sir/Ma for the time, care and effort you have put into reviewing this manuscript. We are grateful for your generous comments.

We gave gone over all the points raised and addressed them. We thank you for pointing out the need to indicate the third ARV used by participants. This has been addressed in result section; page 8, line 160-161.

We have also reflected the confidence intervals in the graph for figures 1-3 as suggested. 

We thank you for spotting the discrepancy in the supplementary figure. This was a typo error; 86 had viral loads and 83 (not 88) did not have viral loads. This correction has been effected in the supplementary figure.

We say thank you for your inputs.

Regards

Editor,

Thank you Sir for your time and efforts. We have addressed the reviewers questions.

We have also uploaded the figures into PACE and figures that meet the PLOS ONE requirements have been uploaded.

We have also uploaded a tracked and clean copy of the manuscript

Thank you Sir in advance for your continuous support.

Best regards

---

## [Editor Report · Decision Letter 2]

10 Aug 2020

The impact of HIV-1 subtypes on virologic and immunologic treatment outcomes at the Lagos university teaching hospital: A longitudinal evaluation

PONE-D-19-29623R2

Dear Dr. Ogbenna,

We’re pleased to inform you that your manuscript has been judged scientifically suitable for publication and will be formally accepted for publication once it meets all outstanding technical requirements.

Kind regards,

Jason Blackard, PhD

Academic Editor

PLOS ONE

Additional Editor Comments (optional):

None

Reviewers' comments:

None

---

## [Editor Report · Acceptance letter]

14 Aug 2020

PONE-D-19-29623R2 

The Impact of HIV-1 Subtypes on Virologic and Immunologic Treatment Outcomes at the Lagos University Teaching Hospital: A longitudinal evaluation 

Dear Dr. Ogbenna:

I'm pleased to inform you that your manuscript has been deemed suitable for publication in PLOS ONE. Congratulations! Your manuscript is now with our production department. 

Kind regards, 

on behalf of

Dr. Jason Blackard 

Academic Editor

PLOS ONE